# Effective Predictor Factors for Lymph Node Metastasis and Survival in Patients with Betel Nut-Related Oral Squamous Cell Carcinoma

**DOI:** 10.3390/diagnostics14222565

**Published:** 2024-11-15

**Authors:** Jiun-Sheng Lin, Yih-Shan Lai, Chieh-Yuan Cheng, Chung-Ji Liu

**Affiliations:** 1Department of Oral and Maxillofacial Surgery, Mackay Memorial Hospital, Taipei 10449, Taiwan; shengd85@yahoo.com.tw (J.-S.L.); phoenix30916@gmail.com (Y.-S.L.); 4715cheng@gmail.com (C.-Y.C.); 2Institute of Oral Biology, School of Dentistry, National Yang-Ming University, Taipei 11221, Taiwan

**Keywords:** tumor thickness, lymph node density, lymph node metastasis level, oral cancer, survival rate

## Abstract

**Background:** The Ministry of Health and Welfare has reported oral cancer to be one of the most prevalent malignant cancers; it has the third highest incidence rate of all cancers and is the fifth leading cause of death among men in Taiwan. Lymph node metastasis in oral cancer usually has a low survival rate, with no significant improvement in the past 30 years. Therefore, a more effective survival predictor is warranted. Many cancer studies have revealed that monitoring tumor thickness and lymph node density, in addition to tumor, node, and metastasis (TNM) stages, can provide more accurate predictions. **Methods:** This retrospective study analyzed data from 612 patients with oral cancer who had the habit of chewing betel nuts. The study focused on tumor thickness, lymph node density, and the regional distribution of lymph node metastasis to determine their effectiveness as predictors. **Results:** The results revealed that a tumor thickness of 6 mm indicated cervical lymph node metastasis and was the optimal cutoff point for overall survival. The optimal cutoff value for lymph node density was 0.04. Patients with a tumor thickness of >6 mm and a lymph node density of >0.04 had significantly lower overall survival rates. Additionally, patients with >1 lymph node metastasis level and lower cervical metastasis exhibited a relatively worse prognosis. **Conclusions:** Therefore, in addition to TNM staging, tumor thickness, lymph node density, and metastasis level are suitable as parameters for predictors that can be used as references for adjuvant therapies for better therapeutic effects.

## 1. Introduction

As one of the most common malignancies occurring in the head and neck, the global epidemiological trends of oral cancer are changing significantly. The estimated number of new cases of oral cancer worldwide increased from 354,864, as reported by GLOBOCAN 2018, to 377,173, according to GLOBOCAN 2020, while the number of new deaths remained stable at approximately 177,000 [1,2]. Oral squamous cell carcinoma is the most prevalent malignant head and neck cancer, characterized by cervical lymph node metastasis. The 2021 cancer registration reports from Taiwan revealed that oral cancer causes >3000 deaths annually, a figure that has not decreased despite advances in medical treatments [3]. Regardless of the progression in surgical techniques and radiotherapy treatment diversity, the 5-year survival rate of oral cancer remains <50%, with particularly low survival rates among patients with lymph node metastasis [4,5]. The seventh American Joint Committee on Cancer (AJCC) staging is most predominantly used for establishing treatment plans and predicting prognosis in Europe and the United States. However, the seventh edition of the AJCC T staging was basically related to the size of the lesion and the involvement of the adjacent structure; this does not indicate the level of tumor invasion or predict lymph node metastasis. Thus, a more effective predictor is warranted in addition to AJCC staging. In the current eighth edition of the American Joint Committee on Cancer (AJCC) TNM classification system, the DOI of the primary tumor has been integrated into the T category for oral cavity squamous cell carcinoma and has shown to be a major element in reframing the staging system. Strictly speaking, tumor thickness differs from DOI, although many clinicians use the two terms interchangeably. DOI is defined as the extent of cancer expansion into the tissue beneath the epithelial basement membrane, while tumor thickness concerns the whole tumor dimension. Truly, DOI requires the final histopathology report to be determined [6,7]. The incidence of occult neck metastasis in non-clinical lymph node metastasis varies from 20% to 30%. Additionally, greater tumor size indicates an increased probability of lymph node metastasis [8,9,10]. Tumor thickness not only represents the tumor size but also demonstrates how the tumor grows in a three-dimensional space, including both endophytic and exophytic growth. As tumor cells grow, the tumor’s capillary proliferation tends to be closer to lymphatic vessels, increasing its likelihood of entering these vessels and the tumor’s metastatic ability, thus making it an indicator of lymph node metastasis [11,12,13]. Han et al. revealed that pathological or radiological examinations of early tongue cancer rarely detect signs of lymph node metastasis [14]. As a result, the patient’s N stage is often underestimated, affecting the overall survival rate [14]. Therefore, a more effective lymph node metastasis predictor is warranted. So far, routine histopathologic examinations have not included lymph node density (number of metastatic lymph nodes/number of removed lymph nodes). Recent studies on bladder and esophageal cancers revealed that the use of lymph node density for survival rate prediction provides better results than traditional TNM staging [15,16], indicating lymph node density as an effective independent predictor. Studies on head and neck cancers have revealed that lymph node density is a reliable predictor of survival rate; however, some analytical data and results are discrepancies [17,18]. In a 2016 study, Suzuki et al. applied an LND value of 0.07 as a predictive factor for lung metastases in OSCC patients [17]. In a study by Ziv Gil et al., data were collected from 186 oral cancer patients and multivariate analysis identified a lymph node density cutoff value of 0.06 as an independent predictor of patient survival rate [18]. An effective predictor should be objective, accessible, and readily comprehensible. For the purpose of identifying effective predictors for lymph node metastasis and overall survival rate, the present study investigated patients from a single medical center with oral squamous cell carcinoma who had the habit of chewing betel nuts.

## 2. Materials and Methods

### 2.1. Patients

This study investigated tumor thickness, lymph node density, and lymph node metastasis area to determine their use as effective predictors. Participants included betel nut-chewing patients from the Department of Oral and Maxillofacial Surgery, Mackay Memorial Hospital, from December 2002 to December 2012. Inclusion criteria were patients with oral squamous cell carcinoma who had a history of betel nut chewing. Patients with a history of betel nut exposure and a betel nut-chewing habit were analyzed according to the MMH oral cancer screening chart. Information on occupation and historical details, such as whether the patients were users (defined as betel quid chewers for under 10 years and less than 20 quids per day, or for more than 10 years and more than 25 quids per day), was collected. Having a betel nut-chewing habit was defined as having consumed at least 10 nuts per day for more than 10 years preceding the first survey appointment and having received surgical treatments, including wide excision of primary oral cancers and neck dissection. Therapeutic neck dissection was performed on all patients with clinically positive nodes (cN+) using a standardized modified radical neck dissection involving levels I–V. Selective neck dissection was performed in patients with clinically negative nodes (cN0), with selective neck dissection involving levels I–III or I–IV. The exclusion criteria were the unclear number of removed lymph nodes, distant metastasis, and having previously received radiotherapy or chemotherapy. Finally, this study included 612 patients and determined patients’ staging based on the AJCC seventh edition. The histopathology reports of these patients were obtained from electronic medical records. Pathological specialists interpreted the records to determine the number of metastatic lymph nodes, removed the lymph nodes, and determined the lymph node metastasis area. Each lymph node was sectioned every 2 mm, put in a different cassette, and embedded in paraffin. Sectioning was performed at 200 μm intervals into the block. LND for each patient was calculated by dividing the number of positive lymph nodes by the total number of lymph nodes removed. Tumor thickness was measured as the vertical depth from the tumor surface to tumor cell invasion in connective tissue through the basement membrane, The tumor maximum value of each dimension (the length, width, and thickness) of the tumors was noted from the final histopathology report.

### 2.2. Statistics

The area under the ROC curve (AUC) was 0.614, sensitivity was 70.91, and specificity was 60.50, indicating that a tumor thickness of 6 mm is the optimal cutoff value for all primary sites of oral cancer (Figure 1A). The AUC was 0.648, sensitivity was 69.32, specificity was 60.42, and lymph node density was 0.04 (Figure 1B). The Kaplan–Meier method was used to construct survival curves, and a univariate Cox regression model was established to estimate the impact of clinicopathological variables on survival. Only variables with a *p*-value < 0.05 were considered statistically significant. Clinicopathological variables that were statistically significant in the univariate analysis were then included in a stepwise forward multivariate Cox regression analysis. Overall survival was defined as the time from the date of surgery to the date of death or the last follow-up. All tests were two-sided with a significance level set at α = 0.05. A *p*-value of <0.05 was considered statistically significant for all analyses.

## 3. Results

This study included 612 patients with oral cancer, including 568 males and 44 females, with an average age of 53.3 years. The buccal mucosa was the most prevalent tumor site, followed by the tongue. Table 1 shows other original data on clinical pathological parameters.

### 3.1. Cox Univariate and Multivariate Analyses of Overall Survival

In the univariate Cox analysis, several clinical factors were identified as significant predictors of overall survival in patients with OSCC (Table 2). These included higher lymph node density (HR 3.99, 95% CI 2.94–5.42, *p* < 0.001), the presence of lymph node metastasis (pN+) (HR 2.96, 95% CI 2.23–3.92, *p* < 0.001), larger tumor size (T4) (HR 3.31, 95% CI 2.31–4.63, *p* < 0.001), and perineural invasion (HR 2.70, 95% CI 2.00–3.67, *p* < 0.001). Additionally, diabetes mellitus (HR 2.59, 95% CI 1.94–3.47, *p* < 0.001), poor cell differentiation (HR 3.03, 95% CI 1.83–5.00, *p* < 0.001), lymphovascular invasion (HR 2.65, 95% CI 1.96–3.60, *p* < 0.001), and increased tumor thickness (>6 mm) (HR 3.89, 95% CI 2.24–6.71, *p* < 0.001) were associated with a significantly increased risk of poor survival. Covariates with a *p*-value < 0.05 in the univariate analysis were further evaluated in the multivariate Cox analysis to assess their independent prognostic significance. Covariates with a *p*-value < 0.05 in the univariate analysis were further assessed using multivariate Cox analysis to determine their independent prognostic value. The multivariate analysis confirmed that higher lymph node density (HR 2.18, 95% CI 1.11–4.28, *p* = 0.024), lymph node metastasis (HR 1.83, 95% CI 1.01–3.34, *p* = 0.048), larger tumor size (HR 2.29, 95% CI 2.16–4.35, *p* < 0.001), the presence of diabetes (HR 2.96, 95% CI 1.71–5.13, *p* < 0.001), perineural invasion (HR 2.32, 95% CI 1.60–3.36, *p* < 0.001), and increased tumor thickness (HR 2.95, 95% CI 1.59–5.47, *p* = 0.001) were independent risk factors associated with decreased overall survival. These findings underscore the importance of these variables in predicting patient prognosis and highlight their potential role in guiding clinical decision-making and treatment strategies for OSCC patients.

### 3.2. Association Among Tumor Thickness, Lymph Node Metastasis, and Overall Survival Rate

A total of 213 (48.6%) and 225 (51.4%) patients were positive and negative for lymph node metastasis, respectively. Figure 2A shows the correlation distribution between tumor thickness and lymph node metastasis. Tumor thickness was ≤6 mm for 44 (20.7%) and >6 mm for 169 (79.3%) of the 213 patients with lymph node metastasis, with a significant difference between them (*p* < 0.001, Table 3). Our results revealed that a tumor thickness of 6 mm is crucial for predicting regional lymph node metastasis. Different tumor sites were further analyzed. A total of 60 (49.6%) and 61 (50.4%) patients were positive and negative for lymph node metastasis in the tongue (9 mm) (*p* < 0.001, Table 3), respectively. Figure 2B shows the correlation distribution between lymph node metastasis and tumor thickness. A total of 89 (50.6%) and 87 (49.4%) patients were positive and negative for lymph node metastasis in the buccal mucosa (7 mm) (*p* < 0.001, Table 3), respectively. Figure 2C shows the correlation distribution between lymph node metastasis and tumor thickness. The Kaplan–Meier curve analysis, comparing patients with a tumor thickness of ≤6 mm to those with a tumor thickness of >6 mm, revealed poorer overall survival among patients with a tumor thickness of >6 mm (*p* < 0.001, Figure 3A). After including tumor thickness in the analysis, patients with lymph node metastasis and a tumor thickness of >6 mm demonstrated poorer overall survival (*p* = 0.006, Figure 3B), indicating tumor thickness as an effective predictor of survival rate.

### 3.3. Association Between Clinical Pathological Parameters and Tumor Thickness

Univariate and multivariate analyses of the clinicopathological parameters’ effects on tumor thickness were performed. The results revealed that perineural invasion (*p* < 0.003), tumor size (*p* < 0.001), pathological stage (*p* < 0.001), and cell differentiation (especially moderate and poor differentiation) (*p* < 0.004) are independent factors affecting tumor thickness (Table 4).

### 3.4. Association Between Lymph Node Density and Overall Survival Rate

Lymph node densities of ≤0.04 and >0.04 were observed in 514 (84.0%) and 98 (16.0%) patients, respectively. Additionally, 143 (59.4%) and 97 (40.6%) patients demonstrated lymph node metastases and lymph node densities of ≤0.04 and >0.04, respectively. Figure 4A shows the correlation distribution between lymph node metastasis and lymph node density. Figure 4B shows the correlation distribution between lymph node metastasis stage and density. The Kaplan–Meier curve analysis, when setting lymph node density 0.04 as the optimal cutoff value, revealed that patients with a lymph node density of >0.04 demonstrated poorer overall survival compared to those with a lymph node density of ≤0.04 (*p* < 0.001, Figure 5A). Patients with lymph node metastasis and density of >0.04 showed poorer overall survival rates when the lymph node density was included for the analysis of lymph node metastasis (*p* < 0.001, Figure 5B). Lymph node density remained a predictor for overall survival rate after including lymph node density in the analysis of the lymph node metastasis stage (*p* < 0.001, Figure 5C).

### 3.5. Association Between Clinical Pathological Parameters and Lymph Node Density

Univariate and multivariate analyses of clinicopathological parameters’ effects on lymph node density were performed. The results revealed lymphovascular invasion (*p* < 0.001), tumor thickness (*p* < 0.044), diabetes mellitus (*p* < 0.011), and cell differentiation, including moderate and poor differentiation (*p* < 0.027) (Table 5), as independent factors affecting lymph node density.

### 3.6. Association Between Lymph Node Metastasis Level and Overall Survival Rate

In the univariate and multivariate Cox analysis of associated mortality according to different levels metastasis of cervical lymph nodes in pN+ patients with OSCC, the level of nodal metastasis was identified as a significant predictor of overall survival in pN+ patients with OSCC (Table 6). Kaplan–Meier curve analysis revealed a poor overall survival rate regardless of lymph node metastasis level (*p* < 0.001, Figure 6A). Further analyses where patients were grouped based on their lymph node metastasis level, all patients whose lymph node metastasis was in the lower neck level (*p* < 0.001, Figure 6B) demonstrated a poorer survival rate.

### 3.7. Association Between Overall Survival Rate and Tumor Thickness Combined with Lymph Node Density

Patients with a lymph node density of >0.04 and a tumor thickness of >6 mm demonstrated poorer overall survival when including lymph node density and tumor thickness simultaneously in survival rate analysis (*p* = 0.005, Figure 7A). Linear regression analysis revealed a positive correlation between tumor thickness and lymph node density (*p* = 0.0053) (Figure 7B). The results of this study confirm that it can be inferred that patients who chew betel nut have larger tumor thickness and that tumor thickness, lymph node density, and lymph node metastasis level are effective predictors of lymph node metastasis and overall survival.

## 4. Discussion

The 7th AJCC TNM staging is a widely used method for cancer prognosis prediction [19]. However, this method often causes errors in predicting the cancer course in some patients, the current cornerstones of therapeutic decision-making, namely the TNM system supplemented with conventional histopathological tumor grading, have proven to be unsatisfactory prognostic indicators. Thus, there is a strong need for further prognostic factor studies focusing mainly on the assessment of the biological aggressiveness of individual tumors underlying clinical malignancy [20,21,22]. Previous studies identified tumor thickness as a predictor for lymph node metastasis [10,13,23]. Dr. O’Brien et al. analyzed 145 cases of T1 and T2 tongue cancer, and the optimal tumor thickness cutoff point was 4 mm [10]. Another study by Dr. Spiro et al. analyzed 92 cases of T1–T2–T3, any N, tongue and floor of mouth cancer, and the study revealed that the optimal tumor thickness cutoff point was 2 mm [13]. In 2005, a review by Pentenero et al. analyzed studies on tumor thickness in predicting regional metastases and survival. The literature suggests that tumor thickness is a reliable parameter for predicting regional nodal involvement and survival in OSCC [23], and the values of tumor thickness ranged from 2 to 10 mm in different studies [24,25]. The significant variation in tumor thickness may be due to unclear tumor thickness definitions and a lack of a standardized measuring method [26]. Many researchers have attempted to determine the thickness value for metastatic capacity, but a consensus has not been reached yet. Giacomara et al. proposed the invasion depth as the tumor range beneath the basement membrane [27]. However, the tumor thickness includes both invasion depth and tumor above the basement membrane and is the measured value of the entire tumor [27]. Breslow was the first researcher to study tumor thickness. He analyzed cutaneous melanoma in 1975 and revealed that different tumor thicknesses affected lymph node metastasis [28]. In 1986, Spiro and Mohit-Tabatabai et al. studied the correlation between oral cancer tumor thickness and lymph node metastasis for the first time [23,29]. Asakage et al. in 1998, Kurokawa et al. in 2002, O-Charoenrat et al. in 2003, and Mücke et al. in 2016 then conducted similar studies, confirming tumor thickness as a predictor for lymph node metastasis [24,25,30,31]. Mohit-Tabatabai et al. conducted a retrospective study that included 84 patients with early-stage oral floor carcinoma and indicated that preventive neck dissection was recommended for cN0 patients with a tumor thickness of >1.5 mm [29]. The retrospective study conducted by Asakage et al. included 44 patients with early-stage tongue cancer. The study revealed 4 mm as the optimal cutoff value, showing statistical significance for lymph node metastasis and disease-free survival rate. Consequently, neck dissection was recommended for early-stage patients with a tumor thickness of >4 mm [30]. Mücke et al. revealed 8 mm as the optimal cutoff value for tongue cancer and recommended using tumor thickness to predict concealed lymph node metastasis [31]. However, Brown et al. and Morton et al. revealed no correlation between tumor thickness and lymph node metastasis [32,33]. The present study revealed that a tumor thickness of 6 mm is the crucial cutoff value for lymph node metastasis, with statistical significance when analyzing the overall survival rate. Moreover, the cutoff values for buccal mucosa and tongue were 7 and 9 mm for different tumor sites, respectively. Our data revealed certain differences from data obtained from overseas studies, which could be attributed to varying cancer risk factors in different countries, tumorigenic sites, and cell differentiation [34]. Patients included in this study were mainly T3–T4 and had the habit of betel nut chewing, with common tumorigenic sites of buccal mucosa and tongue. Betel nut chewing affects tumor thickness as repeated chewing wears out the oral mucosa, and the chronic irritation easily causes exophytic and ulcerative tumor patterns, resulting in tumor proliferation and thickening [35]. Wong et al. applied betel nut extract to hamster cheek pouches and observed increased tumor thickness, indicating that betel nut extract induces cancer and promotes tumor growth [36]. Liao et al. confirmed that cancer cells are mainly presented poorly differentiated cells in countries where betel nut chewing is popular [37]. Huang et al. revealed that moderately and poorly differentiated cancer cells manifest lymph node metastasis [38]. This study compares the risk factors for cancer occurrence and the common anatomical sites of tumors between patients who chew betel nuts and those who do not. Risk factors for oral cancer vary across countries. For example, epidemiological studies have confirmed that betel nut chewing is the primary risk factor in India and Taiwan. In these countries, tumors are most commonly found in the buccal mucosa and tongues of patients who chew betel nuts. However, in Western countries, such as those in Europe and the United States, cancer of the floor of the mouth is the most common site, with patients in these regions generally not chewing betel nuts [31]. According to Liao et al.’s research, it was observed that tumor cells from betel nut-chewing countries tended to show cell poorly differentiation, suggesting that the degree of cell differentiation is an important prognostic factor for patients. In contrast, tumor cells from non-betel nut-chewing countries showed a mostly positive response to moderately differentiated cells and good prognosis [37], which may contribute to differences in key cut-off data for metastasis. These factors partially limit the research results and represent areas that need further investigation and improvement.

The results of the present study revealed that cell differentiation level, especially moderate and poor differentiation, affects tumor thickness. Thus, our research results can be recommended for clinical application in the following ways: 1. Measurements of tumor thickness using preoperative imaging, such as MRI, CT, or sonography, are recommended to determine whether cervical lymph dissection is required. 2. Surgeons can use this as a tool to identify the surgical range of regional lymph nodes for early-stage cancer and clinical N0 patients. For patients with tumor thickness greater than 6 mm, selective neck dissection is strongly recommended.

The key factor affecting cancer prognosis is cervical lymph nodes; however, traditional N staging is not fully capable of prediction. Integrating lymph node density into the traditional TNM staging system could result in more accurate staging and better prognostic assessments, particularly for N+ patients. For example, in oropharyngeal cancer patients with pN1 and pN2 staging, if the number of metastatic lymph nodes exceeds four, but less than four are removed due to inadequate neck dissection, the staging may underestimate the pN1. Therefore, integrating lymph node density into the traditional TNM staging system, especially for lymph node staging, can more effectively predict patient prognosis. Studies on bladder, colorectal, breast, and cervical cancers have identified lymph node density as an effective prognostic predictor [17,39,40,41]. Kim et al. investigated the records of 211 patients with oral cancer to analyze the correlation between lymph node density and survival rate and revealed that patients with a lymph node density of >0.06 demonstrated poorer survival rates [42]. A random-effect model was used for statistical analysis in an international research alliance that included patients with oral cancer from 11 medical centers. This analysis, accounting for data heterogeneity and varying results from individual hospitals, revealed that a lymph node density of >0.06 is usually associated with a poor survival rate of patients, confirming lymph node density as an independent predictor of survival rate [43]. In a retrospective study conducted at the Royal Prince Alfred Hospital Head and Neck Cancer Center in Australia, data were collected from 313 cases of oral cancer. All patients underwent tumor resection and neck dissection. This study analyzed the relationship between lymph node ratio and survival rate, finding that lymph node ratio was a significant prognostic factor for patient survival [44]. Another study applied an LND value of 0.07 as a predictive factor for lung metastases in OSCC patients [17]. In 2009, a study by Ziv Gil and colleagues collected data from 186 oral cancer patients who underwent initial surgical treatment, including neck dissection. The study analyzed the relationship between lymph node density and survival rate. Univariate analysis showed that clinical tumor size, neck lymph node stage, extranodal extension, and lymph node density were factors affecting survival rate. Further multivariate analysis identified a lymph node density cutoff value of 0.06 as an independent predictor of patient survival rate [18].

Some factors may affect the ratio of lymph node density, such as the neck dissection procedure, the interpretive ability of pathologists, and the quality of the histopathologic slide. Inadequate neck dissection will reduce the total number of lymph nodes removed, increase the chance of occult metastasis, and lead to underestimation of the pathological stage. This results in higher lymph node density ratio, ultimately reducing the survival rate of patients [45]. The lymph node density is 0.04 by assuming one patient has one metastatic lymph node (24 total removed lymph nodes), and the lymph node density is 0.2 for another patient with one metastatic lymph node (but 5 total removed lymph nodes). Both patients have the N1 lymph node stage, but those with only five removed lymph nodes need to worry about the risk of occult metastasis and possibly underestimate the N staging.

The advantages of using lymph node density include the following: 1. Consideration of tumor factors: The number of metastatic lymph nodes. 2. Surgical treatment factors: The total number of lymph nodes removed during neck dissection. 3. Tumor staging factors: Pathologists need to carefully identify all lymph nodes and metastases. Another benefit of lymph node density is its usefulness as a quality indicator when performing neck dissection. The total removed lymph node number can be increased to reduce the chance of occult metastasis when including lymph node density as a surgical quality indicator, thereby improving the patient’s survival rate.

The strength of our study lies in the fact that it validated the importance of lymph node density, tumor thickness in a single medical center study that represents a large local population of OSCC patients. The results and detailed statistical analyses demonstrate that lymph node density and tumor thickness can effectively predict patient prognosis. Most patients with advanced oral cancer may require more aggressive adjuvant treatment, including chemotherapy combined with radiation or radiation therapy alone, to improve survival rates. However, some severe side effects can affect the quality of life, reducing patients’ willingness to undergo adjuvant treatment after surgery. There is still controversy over which clinicopathological factors necessitate adjuvant therapy, especially in pN+ patients, who require a more effective evaluation to determine predictive factors for adjuvant treatment. In addition to the traditional TNM staging system, lymph node density serves as a useful adjunct, addressing some of the TNM system’s limitations and acting as an effective factor for determining whether postoperative adjuvant therapy is needed.

The retrospective nature of the study has some limitations. The history of betel nut chewing was based on patient self-reports, which may lead to underreporting. Creating a detailed chart for betel nut chewing habits could allow for further subdivision into mild, moderate, or severe levels, including the number of betel nuts chewed, the duration of chewing habits, and different types of betel nut products. Additionally, it should assess whether betel nut juice was swallowed. Given that carcinogens vary from country to country, our findings have limited generalizability to other countries.

Future research aims to use this large sample size to identify more effective prognostic factors, such as tumor-infiltrating lymphocytes (TILs), which are T cells surrounding the tumor and serve as a marker of the host’s response to the tumor. By analyzing the depth of tumor invasion and TILs in tissue sections, we can investigate correlations. OSCCs of different sites have differences in etiology. SCC of FOM appears to be almost entirely cancer caused by the common carcinogens of tobacco and alcohol. Otherwise, SCC of buccal appears to be almost entirely cancer caused by the betel nut chewing. Different carcinogens induce oral cancer in different anatomical locations. Molecular expression profiles of these oral cancers associated with endogenous factors also warrant investigation.

This study revealed a lymph node density of 0.04 as the optimal cutoff value. Patients with pN+ and a lymph node density greater than 0.04 demonstrated a poorer prognosis. Therefore, based on previous studies and our findings, integrating lymph node density into the traditional N staging would allow more accurate prediction of patient prognosis. Moreover, the overall survival rate was poor when more than one metastasis level was observed or when the metastasis was in the lower neck level.

## 5. Conclusions

The advantages of this study are that all participants were from the same medical center and that >600 oral cancer patients with the habit of betel nut chewing underwent the same type of neck dissections based on therapeutic standards, thereby adequately representing sufficient local samples in Taiwan. Our results indicated that lymph node density, tumor thickness, and regional distribution of lymph node metastases are effective prognostic predictors that can be used as references for adjuvant therapies for the better overall survival of patients. Future studies are warranted that use a larger sample size to determine more effective predictors. For example, the correlation between tumor-infiltrating lymphocytes and the depth of tumor invasion can be analyzed.

## Figures and Tables

**Figure 1 diagnostics-14-02565-f001:**
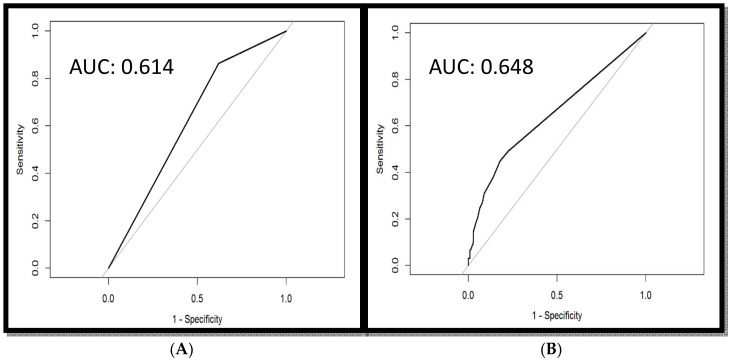
(**A**) The receiver operating characteristic curve for the tumor thickness value. The tumor thickness value of 6 mm is the optimal cutoff value. (**B**) The receiver operating characteristic curve for the lymph node density ratio. The lymph node density ratio of 0.04 is the optimal cutoff value.

**Figure 2 diagnostics-14-02565-f002:**
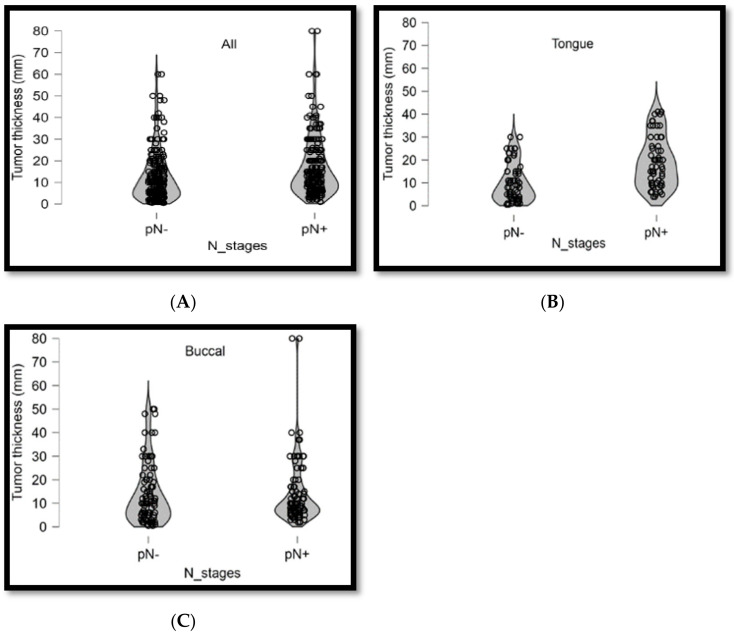
(**A**) Correlation distributions between tumor thickness and lymph node metastasis for cancers of all anatomic sites. (**B**,**C**) Correlation distributions between tumor thickness and lymph node metastasis for cancers of different anatomic sites.

**Figure 3 diagnostics-14-02565-f003:**
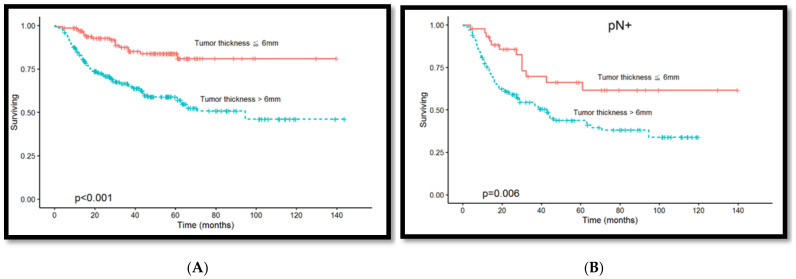
(**A**) Kaplan–Meier survival rate curve; comparison between patients with a tumor thickness of >6 mm and ≤6 mm, where patients with a tumor thickness of >6 mm demonstrated poorer overall survival (*p* < 0.001). (**B**) Patients with lymph node metastasis and tumor thickness of >6 mm demonstrated poorer overall survival rates after including tumor thickness in survival rate analysis (*p* = 0.006).

**Figure 4 diagnostics-14-02565-f004:**
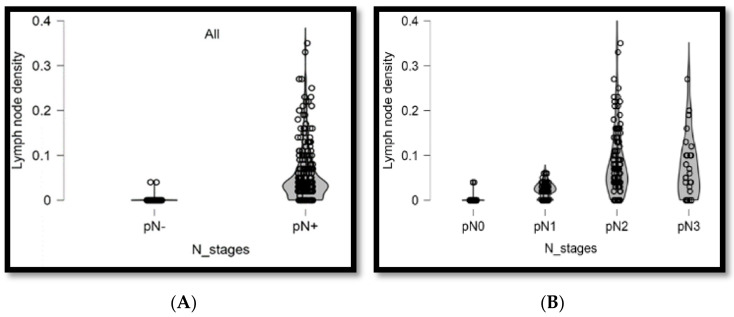
(**A**,**B**) Correlation distribution diagram between patients with lymph node metastasis and lymph node density. Lymph node metastasis cases are divided into pN1, pN2, and pN3 stages. Correlation distribution diagram between lymph node metastasis stage and lymph node density.

**Figure 5 diagnostics-14-02565-f005:**
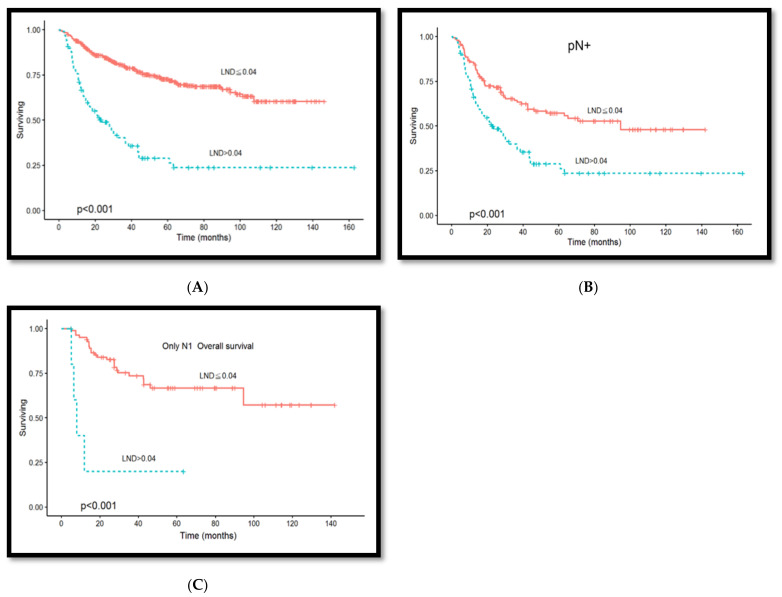
(**A**) Kaplan–Meier survival rate curve; comparison between patients with lymph node density of >0.04 and ≤0.04, where patients with a lymph node density of >0.04 exhibited poorer overall survival (*p* < 0.001). (**B**) Patients with lymph node metastasis and lymph node density of >0.04 demonstrated a poorer overall survival rate after including lymph node density in survival rate analysis (*p* < 0.001). (**C**) Patients in the pN1 stage with a lymph node density of >0.04 exhibited poorer overall survival rates; we divided the patients with lymph node metastasis into the pN1, pN2, and pN3 groups and included lymph node density into the survival rate analysis of patients with different lymph node metastasis stages (*p* < 0.001). No significant difference was found between patients with pN2 and those with pN3.

**Figure 6 diagnostics-14-02565-f006:**
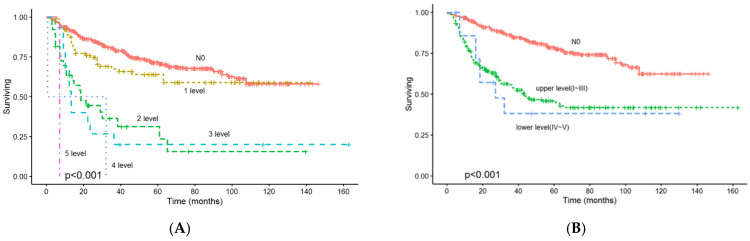
(**A**) Kaplan–Meier lymph node metastasis level distribution curve and overall survival rate. The results revealed poor overall survival rates, regardless of lymph node metastasis level (*p* < 0.001). (**B**) Further analysis revealed that patients with lymph node metastasis in lower cervical level (level VI–V) (*p* < 0.001) demonstrated poorer survival rates.

**Figure 7 diagnostics-14-02565-f007:**
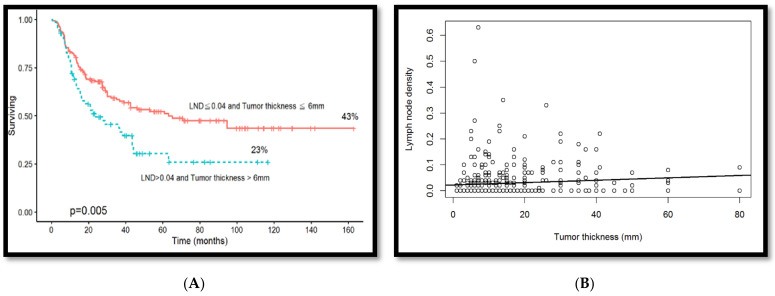
(**A**) After including both lymph node density and tumor thickness for survival rate analysis, patients with a lymph node density of >0.04 and tumor thickness of >6 mm demonstrated poorer overall survival rates (*p* = 0.005). (**B**) Linear regression was utilized to examine the correlation between the overall survival rate and the two parameters of tumor thickness and lymph node density (*p* = 0.0053).

**Table 1 diagnostics-14-02565-t001:** Clinicopathological characteristics of the study participants.

Characteristics		No. of Patients (%)
Gender	Female	44 (7.2%)
	Male	568 (92.8%)
Age	Mean	53.3 y/o
DM	No	486 (79.4%)
	Yes	126 (20.6%)
Tumor size	T1~T3	259 (42.3%)
	T4	353 (57.7%)
Lymph node stage	pN0	372 (60.8%)
	pN+	240 (39.2%)
Pathologic stage	I–III	583 (95.3%)
	IV	29 (4.7%)
Primary site	Buccal	246 (40.2%)
	Tongue	170 (27.7%)
	Other	206 (32.1%)
Cell differentiation	Well	219 (35.8%)
	Moderate + poor	393 (64.2%)
Lymphovascular invasion	No	500 (81.7)
	Yes	112 (18.3%)
Perineural invasion	No	479 (78.3%)
	Yes	133 (21.7%)
Extranodal extension	No	522 (89.7%)
	Yes	60 (10.3%)
Tumor thickness	≤6 mm	133 (30.5%)
	>6 mm	305 (69.5%)
Lymph node density	≤0.04	514 (84.0%)
	>0.04	98 (16.0%)
Level of nodal metastasis	Level I~III	223 (92.9%)
	Level IV~V	17 (7.1%)
Survival status	Alive	414 (67.7%)
	Dead	198 (32.4%)
Follow-up	Mean	54.5 months

y/o: Year old.

**Table 2 diagnostics-14-02565-t002:** Univariate and multivariate analyses of overall survival in patients with OSCC.

Variable	Univariate Analysis		Multivariate Analysis	
	HR (95%CI)	*p*-Value	HR (95%CI)	*p*-Value
Lymph node density				
≤0.04 vs. >0.04	3.99 (2.94–5.42)	<0.001	2.18 (1.11–4.28)	0.024
Lymph node status				
pN0 vs. pN+	2.96 (2.23–3.92)	<0.001	1.83 (1.01–3.34)	0.048
Tumor size				
T1–3 vs. T4	3.31 (2.31–4.63)	<0.001	2.29 (2.16–4.35)	<0.001
Diabetes mellitus				
No vs. yes	2.59 (1.94–3.47)	<0.001	2.96 (1.71–5.13)	<0.001
Gender				
Female vs. male	1.46 (0.89–2.40)	0.138		
Cell differentiation				
Well vs. moderate + poor	3.03 (1.83–5.00)	<0.001		
Perineural invasion				
No vs. yes	2.70 (2.00–3.67)	<0.001	2.32 (1.60–3.36)	<0.001
Lymphovascular invasion				
No vs. Yes	2.65 (1.96–3.60)	<0.001		
Tumor thickness				
≤6 mm vs. >6 mm	3.89 (2.24–6.71)	<0.001	2.95 (1.59–5.47)	0.001

**Table 3 diagnostics-14-02565-t003:** Association between tumor thickness at different tumor sites and lymph node metastasis.

Primary Site		pN0 (%)	pN+ (%)	*p*-Value
All	Tumor thickness ≤ 6 mm	89 (39.6%)	44 (20.7%)	<0.001
	Tumor thickness > 6 mm	136 (60.4%)	169 (79.3%)	
Tongue	Tumor thickness ≤ 9 mm	33 (54.1%)	12 (20.0%)	<0.001
	Tumor thickness > 9 mm	28 (45.9%)	48 (80.0%)	
Buccal	Tumor thickness ≤ 7 mm	35 (40.2%)	23 (25.8%)	<0.001
	Tumor thickness > 7 mm	52 (59.8%)	66 (74.2%)	

**Table 4 diagnostics-14-02565-t004:** Univariate and multivariate analyses of clinicopathological parameters’ effects on tumor thickness.

Variables	Univariate		Multivariate	
	HR (95% CI)	*p*-Value	HR (95% CI)	*p*-Value
Lymphovascular invasion				
No	Reference			
Yes	2.97 (1.64–5.39)	<0.001		
Perineural invasion				
No	Reference		Reference	
Yes	3.99 (2.18–7.31)	<0.001	2.75 (1.41–5.36)	<0.003
Tumor size				
T1–T3	Reference		Reference	
T4	8.25 (5.12–13.28)	<0.001	2.55 (1.13–5.74)	<0.001
Pathologic stage				
I–III	Reference		Reference	
IV	1.75 (1.37–2.22)	<0.001	4.13 (1.87–9.11)	<0.001
Level of nodal metastasis				
Level I–III	Reference			
Level IV–V	1.06 (0.26–4.23)	0.929		
Involved level				
1 Level	Reference			
2 Levels	1.39 (0.59–3.25)	0.442		
>2 Levels	2.22 (0.47–10.31)	0.308		
Extranodal extension				
No	Reference			
Yes	3.68 (1.62–8.35)	0.002		
Diabetes mellitus				
No	Reference			
Yes	2.40 (1.32–4.38)	0.004		
Cell differentation				
Well	Reference		Reference	
Moderate + poor	1.94 (1.27–2.97)	<0.002	2.08 (1.25–3.467)	<0.004

**Table 5 diagnostics-14-02565-t005:** Univariate and multivariate analyses of clinicopathological parameters’ effects on lymph node density.

Variables	Univariate		Multivariate	
	HR (95% CI)	*p*-Value	HR (95% CI)	*p*-Value
Lymphovascular invasion				
N0	Reference		Reference	
Yes	6.84 (4.24–11.01)	<0.001	4.84 (2.58–8.22)	<0.001
Perineural invasion				
No	Reference			
Yes	3.29 (2.08–5.22)	<0.001		
Tumor size				
T1–T3	Reference			
T4	1.61 (1.02–2.548)	0.041		
Pathologic stage				
I–III	Reference			
IV	1.57 (0.95–2.58)	0.073		
Level of nodal metastasis				
Level I–III	Reference			
Level IV–V	2.28 (0.79–6.60)	0.126		
Involved level				
1 Level	Reference			
2 Levels	5.98 (3.10–11.53)	<0.001		
>2 Levels	9.28 (3.34–25.72)	<0.001		
Extranodal extension				
No	Reference			
Yes	17.87 (9.75–32.76)	<0.001		
Tumor thickness				
≤6 mm	Reference		Reference	
>6 mm	2.92 (1.56–5.49)	<0.001	1.98 (1.02–3.86)	<0.044
Diabetes mellitus				
No	Reference		Reference	
Yes	2.16 (1.34–3.49)	<0.002	2.09 (1.18–3.86)	<0.011
Differentiation				
Well	Reference		Reference	
Moderate + poor	2.46 (1.46–4.15)	<0.001	1.987 (1.08–3.64)	<0.027

**Table 6 diagnostics-14-02565-t006:** Analyses of associated mortality according to different levels metastasis of cervical lymph nodes in pN+ patients.

Variables	Univariate		Multivariate	
	HR (95% CI)	*p*-Value	HR (95% CI)	*p*-Value
Level of nodal metastasis				
I–III	Reference		Reference	
IV–V	1.87 (1.06–3.30)	0.03	0.11 (0.03–8.22)	0.003
Involved level				
1 Level	Reference			
2 Levels	2.22 (1.46–3.38)	<0.001		
>2 Levels	2.23 (1.27–3.83)	0.01		

## Data Availability

No new data were created or analyzed in this study. Data sharing is not applicable to this article.

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
