# Peer review of "Effective Predictor Factors for Lymph Node Metastasis and Survival in Patients with Betel Nut-Related Oral Squamous Cell Carcinoma"

_diagnostics, 2024, doi:10.3390/diagnostics14222565_

Round 1

Reviewer 1 Report

Comments and Suggestions for Authors

Dear Author, 

Please find the comments document attached

Regards

Reviewer 2 Report

Comments and Suggestions for Authors

The research article submitted for review has some aspects that needs to be addressed. Hence suggesting major revision.

Abstract: Adequate

Introduction

1.      P1L42: The author mentioned the 2008 version of the AJCC, which is incorrect. Depth of invasion in oral cavity cancer was included in the 2018 version (8th edition). Additionally, it also differentiated and defined the terminologies of tumor thickness and DOI and preference was given to the later. I would like to suggest that the article including results be updated according to the definition of 8th edition.

a. Amin MB, Edge SB, Greene FL, et al, eds.AJCC Cancer Staging Manual. 8th ed. NewYork: Springer; 2017. 

b. Lydiatt WM, Patel SG, O'Sullivan B, Brandwein MS, Ridge JA, Migliacci JC, Loomis AM, Shah JP. Head and Neck cancers-major changes in the American Joint Committee on cancer eighth edition cancer staging manual. CA Cancer J Clin. 2017 Mar;67(2):122-137. doi: 10.3322/caac.21389. Epub 2017 Jan 27. PMID: 28128848.)

  1. P2L55: There is no article by Han et al. in the references.

Materials and Methods

  1. P2L77: 10y?
  1. P2L81: Why were patients not categorized according to the 8th edition? As DOI is preferred because it can provide standardized measurement for both exophytic and endophytic growths. The author has consistently used tumor thickness. Why was the research not conducted according to the recent AJCC guidelines and methodology recommending DOI for analysis?

a.       Lydiatt WM, Patel SG, O'Sullivan B, Brandwein MS, Ridge JA, Migliacci JC, Loomis AM, Shah JP. Head and Neck cancers-major changes in the American Joint Committee on cancer eighth edition cancer staging manual. CA Cancer J Clin. 2017 Mar;67(2):122-137.

b.      Lee NCJ, Eskander A, Park HS, Mehra S, Burtness BA, Husain Z. Pathologic staging changes in oral cavity squamous cell carcinoma: Stage migration and implications for adjuvant treatment. Cancer. 2019 Sep 1;125(17):2975-2983.)

Results

  1. Table 1: DM. Diabetes Mellitus or Distant Metastasis?

Discussion

  1. P11L276: Should be Liao et al.
  2. The gold standard for measuring DOI and tumor thickness is histopathology. The author should specify how tumor thickness was measured in this study in all the cases. Why has the author not compared the MRI, CT, and ultrasonography results with histopathology and compared sensitivity and specificity since lesions with a DOI of about 4mm may not be distinguishable through CT and MRI. (Fernanda Marcello Scotti, Rubia Teodoro Stuepp, Kamile Leonardi Dutra-Horstmann, Filipe Modolo, Marcelo Gusmão Paraiso Cavalcanti, Accuracy of MRI, CT, and Ultrasound imaging on thickness and depth of oral primary carcinomas invasion: a systematic review, Dentomaxillofacial Radiology, Volume 51, Issue 5, 1 July 2022, 20210291.)
  3. et al. is missing from the names of referenced author.
  4. What were the limitations of the study?

References: Adequate

Comments on the Quality of English Language

Satisfactory

Reviewer 3 Report

Comments and Suggestions for Authors

INTRODUCTION

The introduction discusses the  of oral cancer in Taiwan, but it could benefit from a more global perspective.

METHODS

The inclusion criteria are generally well-defined, but the details on the quantity of Betel Nut consumption (10–25 quids daily for 10 years) need further explanation. What is the justification for this specific range? Was it validated in previous studies?

DISCUSSION

The discussion is well organised, but it does not adequately explain how the study advances the field. More comparison with existing literature on lymph node density and survival prediction in oral cancer is needed.

How does Betel Nut affect tumor biology differently from other known risk factors like tobacco or alcohol? 

Round 2

Reviewer 2 Report

Comments and Suggestions for Authors

The autor has revised the article and addressed the concerns.